# The Role of Focal Therapy and Active Surveillance for Small Renal Mass Therapy

**DOI:** 10.3390/biomedicines10102583

**Published:** 2022-10-14

**Authors:** Milena Matuszczak, Adam Kiljańczyk, Maciej Salagierski

**Affiliations:** Department of Urology, Collegium Medicum, University of Zielona Góra, 65-046 Zielona Góra, Poland

**Keywords:** focal therapy, kidney cancer, thermal ablation

## Abstract

Small and low-grade renal cell carcinomas have little potential for metastasis and disease-related mortality. As a consequence, the main problem remains the use of appropriately tailored treatment for each individual patient. Surgery still remains the gold standard, but many clinicians are questioning this approach and present the advantages of focal therapy. The choice of treatment regimen remains a matter of debate. This article summarizes the current treatment options in the management of small renal masses.

## 1. Introduction

The detection rate of small renal masses (SRMs) is increasing every year. This is mainly due to improvements in diagnostic methods, as well as increased life expectancy, which contributes to the possibility of recurrence. In 2020, there were 431,288 kidney cancer cases and 179,368 deaths worldwide [1]. The estimated number of cases in the United States in 2022 is 79,000, and that of deaths is 13,920 [2]. Kidney cancer is most often detected incidentally when imaging is performed for other reasons and occurs about 2 times more often in men than in women. More than half of currently diagnosed renal masses are detected incidentally [3]. SRMs are defined by most of the literature as smaller than 4 cm, which is usually synonymous with grade T1a of the TNM classification of renal cell carcinoma (RCC) [4]. The typical triad of symptoms (hematuria/abdominal mass/flank pain) is rarely seen nowadays, as it is associated with advanced RCC, which is diagnosed less and less frequently. The specific survival rate for T1-T2 stage RCC is as high as 80–90% after 5 years [3]. Scientific data prove that small and low-grade renal cell carcinomas have little potential for metastasis and disease-related mortality. The main problem in their therapy is the use of appropriately tailored treatment for the patient. Among the therapeutic approaches, we distinguish active surveillance (AS), partial nephrectomy (PN), radical nephrectomy (RN), focal therapy (FT) in which cryoablation, radiofrequency ablation (RFA), microwave ablation (MWA), and irreversible electroporation (IRE) are included. Surgery still remains the gold standard, but many clinicians are questioning it and presenting the advantages of alternative methods such as FT or AS. Which method to choose and what treatment regimen to use still remain matters to consider and question. This manuscript, which analyzes data from the last 4 years in this area, aims to answer at least briefly the above-mentioned questions, pointing out the advantages and disadvantages of each solution.

## 2. Methods

A literature review was performed by searching PubMed/MEDLINE database from August 2018 to July 2022 to identify studies on the role of the focal therapy (FT) and active surveillance (AS) of small renal masses (SRM). The search terms included small renal masses; renal cell carcinoma; renal cancer; kidney neoplasm; focal therapy; active surveillance; cryoablation; thermoablation; microwave ablation; radiofrequency using search terms database = specific—medical subject headings terms in various combinations appropriate to the research objective.

Papers presenting data in the form of reviews, letters to the editor, editorials, research protocols, case reports, brief correspondence and articles not published in English were excluded. Co-workers checked the literature of all included papers for additional studies of interest. On this basis, articles published before April 2018 were also included (17 articles).

Publications based on tissue, blood, cell lines and animals were excluded. Articles concerning more than one cancer, e.g., additional prostate or bladder were omitted. In addition, papers focusing on technical feasibility and specifications of measurement methods rather than method and clinical utility were excluded. Publications based on small cohorts, i.e., including fewer than five patients, were also excluded.

Researchers independently extracted the following information from the included articles: author name, year of publication, number of patients, stage and/or grade of cancer, tumor size (mean and/or median), follow-up time (mean and/or median) as well as OS, RFS, DSS, MFS, DFS, PFS, PE, SE to assess oncological outcomes. All data extraction discrepancies were resolved by consensus with the co-authors.

## 3. Results

### 3.1. Thermal Ablation

Thermal ablation (TA) involves the destruction of tumor cancer cells using extreme temperatures (both high and low) by one or more applicators. This method includes RFA, MWA, IRE and CA. Of these, RFA and CA appear to be the most studied [5]. These approaches are the most widely used, have the best long-term results and similar oncologic efficacy with no significant differences in OS, CSS and RFS [6].

The European Association of Urology (EAU) [7] gives a (weak) recommendation that this technique should be offered on an equal basis with AS to SRM patients with poor health and/or comorbidities, but stresses that one should always remember to discuss the potential benefits and risks, as well as the possible complications and oncologic effect of the chosen therapeutic option (strong recommendation). The publications prove that there is no significant difference between 5-year CSS with AS or TA [8]. 

The American Urological Association (AUA) and the National Comprehensive Cancer Network (NCCN) [5,9] recommend TA as an alternative treatment option for cT1a tumors <3 cm in size. RFA as well as CA provide similar oncologic outcomes and can therefore be used as an option when choosing TA. These interventions can also be used for larger lesions however, recommendations [9] mention that this approach is associated with higher recurrence rates and more frequent complications.

In addition, the NCCN authors based on AUA 2017 [10] and Pierorazio et al. [11] warn that ablative techniques may require multiple approaches to achieve an oncologic outcome similar to conventional surgery. They recommend using a percutaneous technique whenever possible because of the reduced mortality rate.

#### 3.1.1. Cryoablation

Cryoablation can be performed either laparoscopically or percutaneously, both techniques have a success rate of over 95%. Despite this, recommendations advise (NCCN) to use the percutaneous approach. Although cryoablation is classified as a thermal ablation method, the EAU guidelines [7] in contrast to the group-wide restriction against using TA for tumors over 3 cm, set a slightly higher limit for CA—for tumors over 4 cm.

Compared to standard surgery, CA provides similar performance in terms of disease-free survival (DFS), in addition, it is associated with lower complication rates, but carries the risk of more frequent tumor recurrence [12,13]. His trend increases with clinical stage as demonstrated by a cohort of 308 patients [14]. For cT1a tumors, the recurrence rate was 7.7% compared to cT1b where the rate rose as high as 34.5%. Other publications have reported values for cT1: RFS = 93.9%, MFS = 94.4% [15] and 10-year DFS = 94% [16].

It was indicated that laparoscopic CA was significantly associated with better preservation of renal function at month 6 compared to PN [17]. The authors [18] suggest that there is no significant difference between RFA and cryoablation in recurrence rate, metastatic progression, incidence of complications or length of CSS. In addition, a more recent publication [19] reported that recurrence occurs less frequently after cryoablation than after RFA. The method is also safe and effective for senior patients. A publication [20] proved that the procedure is easy to perform, has a low complication rate and is well tolerated by the elderly.

A recently published paper [13] confirms previous reports that this method preserves renal function and does not lead to a significant decline in its function after treatment. Their conclusions are based not only on the available literature but also on the basis of the cohort studied. In addition, the authors remind us that this method is performed percutaneously and very rarely requires general anesthesia, which saves time, money, as well as prevents potential side effects of anesthesiology. It is worth mentioning that possible reports of complications may be related to the eligibility of patients who are disqualified from more invasive methods due to high ASA score.

A publication [12] reported that repeat cryoablation has a significantly lower success rate compared to the original procedure. In most cases, failure after CA can be repaired with re-cryoablation, but this is the point at which it is worth considering other alternatives. Re-cryoablation unfortunately achieves poor results and only 45% of patients have a 2-year DFS. In terms of cryoablation, CT shows a significant advantage over laparoscopic or navigated ultrasound approaches.

In summary, we can mention the unfavorable risk of recurrence compared to standard surgery, as well as the poor results of re-cryoablation, but this method has many advantages in terms of 3- and 5-year OS, low complication rate and avoids general anesthesia. The potential disadvantage of what may be the appearance of recurrence is not significant, and the technique provides good MFS.

#### 3.1.2. Radiofrequency Ablation

RFA is a technique that uses radiofrequency energy delivered through a needle inserted into a cancerous tumor, causing necrosis of the tissue. The method was first described in 1997 [21], and for many years has ranked as a recommended method in therapy for those who are not in sufficient condition for surgery, and the tumor appears to be able to be completely cured by ablation.

The US FDA has approved a method of high-temperature ablation of soft tissue tumors such as SRM of the kidney. The effectiveness of this method has been described in extensive research [22]. 

RFA is a safe and effective method for the treatment of SRM less than 3 cm in diameter, with therapeutic success in up to 97% of patients. Patients with such tumors as reported in the study [22] had relatively good oncologic outcomes and a 10-year survival rate, with DFS of 82%, CSS of 94% and OS of 49%. No recurrences developed at 5 years after intervention, but patients with tumors larger than 3 cm had worse outcomes (10-year DFS = 68%).

RFA, like cryoablation, is associated with more frequent recurrences than surgical procedures. The technique offers satisfactory results, especially in an aging population due to the reduction in mortality, recovery time, and risks that classical surgery poses.

#### 3.1.3. Microwave Ablation

This technique is categorized among other alternatives. The EAU has not made any recommendations.

In an article [23], researchers described the effectiveness of using percutaneous microwave ablation (MWA) in renal cell carcinoma (RCC) of T1 stage. It is worth mentioning that their cohort included not only SRM, but also larger tumors, but smaller than 7 cm (with an average tumor size of 3.2 cm). They analyzed populations of 100 patients (108 tumors) undergoing treatment over 6 years. Unsurprisingly, the group of patients with T1a tumors achieved better results from treatment compared to T1b. Primary efficacy was 89% and 52%, for T1a and T1b, respectively. Fifteen lesions (including 7 T1a) underwent MWA reablation for residual disease in one (*n* = 13) and two (*n* = 2, both T1b) sessions, achieving secondary efficacy rates of 99% (T1a) and 95% (T1b). Local tumor recurrence (LTR) was equally frequent in both groups (2 each for T1a and T1b). Adverse effects (clinically significant ones were included—grade 3–5 of the Clavien–Dindo classification) were 2 times more frequent in the T1b group than in T1a (2 T1a and 4 T1b were described). Based on the above results, it can be concluded that MWA is a safe treatment option for RCC in both T1a and T1b stages (however, this approach is less effective in more advanced tumor).

A report [24] based on the observation of 48 patients with RCC (with a mean size of 3.1 cm) showed that this method achieves satisfactory OS (95.8%) with few non-significant (observation of hematomas in 4% of patients) complications. One of the disadvantages of this method reported in the literature is the frequency of recurrence, however, nowadays more and more publications show the low severity of the magnitude of this problem (6.25% described here) with increasingly better clinical successes (97.9% overall). Therefore, the authors of the study concluded that this method is an effective technique for SRM and medium-sized tumors.

In addition, the 2021 paper [25] based on a cohort of 101 patients confirms these reports—MWA is a safe and effective (Table 1) treatment for SRM, with a low relapse rate and minimal side effects. However, the authors note the need to observe long-term outcomes.

#### 3.1.4. Irreversible Electroporation

Irreversible electroporation (IRE) is a new non-thermal focal ablation technique that uses a series of short but intense electrical pulses delivered through paired electrodes to the targeted tissue area, killing cells by irreversibly disrupting the integrity of the cell membrane. The effect of IRE is not uniform and depends on the internal conductivity of the tissue, the number of pulses delivered, the current flow achieved and the total treatment time. In clinical practice, it can be performed both percutaneously under imaging guidance (e.g., CT) and during open surgery under direct visual guidance. IRE is a less invasive method for the patient than other ablations, due to its low impact on nerves or connective tissue. This makes the method more suitable for tumors located in the area of vital large vessels, as it allows the lesion to be removed without damaging them. Its low invasiveness also argues for performing procedures using it in patients in severe general condition, with comorbidities or during treatment with chemotherapeutic agents. 

We prospectively evaluated CT [37]—it navigated IRE and showed suboptimal results and acceptable complications (Table 1). Thirty RCC tumors with an average size of 2.5 cm were treated with this method and achieved a primary technique success rate of 73.3%, which improved to 97% after performing CA-7 residual disease. However, it should be mentioned that so far this method is insufficiently studied and carries a high risk of complications (one patient had a complication of Clavien–Dindo III—damage to the proximal ureter and five patients had a decrease in eGFR of more than 25% immediately after IRE). However, all patients had sufficiently well-preserved renal function that they did not require dialysis. One patient did not have a repeat procedure, as he died of an unexpected stroke at 4 months after IRE.

### 3.2. Active Surveillance

Over the past few years, active surveillance has been the recommended treatment option for patients with tumors less than 2 cm in diameter. Such management is based on studies showing that many tumors less than 2 cm required no intervention, and that the delayed interventions used did not differ in terms of metastasis or mortality [6].

To verify which patients with SRM will benefit more from Robot-assisted laparoscopic partial nephrectomy (partial nephrectomy is currently the preferred surgical strategy due to preservation of renal function and excellent oncologic outcomes) vs. AS, a review was conducted [39] of the Delayed Intervention and Surveillance of Small Kidney Masses (DISSRM) Registry data collected over 10 years. This registry includes patients with cT1a tumors <4 cm (in the axial dimension of imaging), after exclusion of familial RCC syndromes and metastases. This work was created with the idea of eliminating unnecessary surgeries, as data show that approximately 5624 needless resections of benign SRMs are performed annually in the US [40].

The risk of using AS is the appearance of metastases, however, their incidence is less than 1% for tumors <3 cm in diameter and about 2% for tumors <4 cm [41,42]. The low rates of metastatic progression (1–6% of literature reports) and SRM-related mortality (0–18%) for untreated small RCC support the choice of this treatment modality. 

AS currently has various recommendations from medical societies:

NCCN recommends AS as an option for selected asymptomatic T1 patients:-with SRM < 2 cm,-with T1a tumors (≤4 cm) with a predominantly cystic component.-with cT1 SRM and significant competing risks of death or morbidity associated with the intervention.

According to the NCCN definition, AS includes:-serial abdominal imaging studies-periodic blood tests and chest imaging (verification of possible metastases)-interventions in a timely manner if the mass shows changes indicative of progression (e.g., increasing tumor size, rapid growth, infiltration) indicating increasing metastatic potential.

The American Society of Clinical Oncology (ASCO) recommends its use as an initial treatment for populations with significant comorbidities and poor predicted survival [4].

AUA, on the other hand, suggests the initial use of this method, for any patient with a tumor less than 2 cm or for larger lesions in an elderly and ailing population, [5] as well as for those at high risk of complications from surgical intervention. AS is an option that requires careful clinical risk assessment, patient and physician co-decision-making, and periodic reevaluation (reassessment). Post-intervention follow-up allows identification of potential implications of treatment and local or systemic recurrence. No consensus has been established on the exact timing of imaging study surveillance.

EAU guidelines [7] recommend AS as an initial method of monitoring SRM, which can always be changed to another therapeutic method. The recommendations for this strategy are mainly for the aged and sickly, who could suffer more losses from more invasive methods, and those whose life expectancy is low. Besides, this method can also be considered in other patients, due to the fact that the 5-year follow-up showed no significant difference in CSS between the AS group and surgical patients [43]. Based on the results of the biopsy and the determination of its histological specification, it is possible to assess the risk of progression and whether the tumor can be safely overseen or whether more invasive methods should be undertaken [39].

Factors triggering intervention in the DISSRM registry include tumor size (>4 cm), growth rate (>0.5 cm/year) [44], symptom development (hematuria with no other cause), elective change (change in patient preference or improvement in patient health), or metastatic disease.

In addition, Mir et al. [45] showed that the linear growth rate of patients who developed metastases was not significantly different from the overall growth rate of clinically localized renal masses. Moreover, because both benign and malignant lesions can grow at similar or non-zero rates, growth rate thresholds alone should not be used as a predictor of mass histology or malignancy potential.

Cancer-specific death and progression of metastatic disease do not appear to be related to the rate of tumor growth [46]. Post-treatment based on overall tumor size is now recommended, as it has been shown to be the best predictor of malignant histology, aggressive pathology and oncologic outcomes [47].

## 4. Discussion 

In our review paper, we collected results from 24 papers on SRM ablative techniques. In total, the data cover 2150 patients at stage cT1a or cT1b, and includes more advanced tumors than previous reports [48].

Considering the characteristics of radicality with which surgical treatment is associated, and because it is undeniably the longest follow-up (period of observation for this method), it remains the standard of care (SOC) for SRM and localized RCC. An alternative is FA, which for several years now has been an officially accepted method of treating SRM with efficacy similar to PN (for tumors <3 cm). The results achieved with FT are satisfactory, and moreover, percutaneous TA carries a lower risk of serious complications than even minimally invasive surgery.

Among TA, the available literature suggests the superiority of CA over RFA in terms of local tumor control and less frequent reoperations (retreatments). However, there are skeptical voices from researchers who question the necessity of intervention due to the low malignancy and risks that are associated with the natural course of SRM. Due to the development of accessibility and the possibility of regular imaging studies, it is increasingly recommended that AS be undertaken in (high-risk) patients with contraindications to surgery. However, all agree that before making a decision, it is important to consider all variables that may affect the patient’s health and also to assess whether the intervention is beneficial to the patient. Published data in the literature indicate that AS is a safe intervention, and TA in the elderly should only be undertaken when outweighed by the gains made during AS.

Mean results for CA, RFA, MWA and IRE procedures are summarized in Table 2.

The average 5-year survival results for CA were 87.97%, 94.08%, 97.96%, 98.6% for OS, RFS, DSS and MFS for cT1a tumors, respectively. For cT1b, the results were: 77.78%, 92.68%, 93.8%, 93.7% for OS, RFS, DSS and MFS, respectively. Which, when compared to the work of Aron et al. 2010 [49] with OS, RFS, DSS and MFS results of 84%, 87%, 89% and 89% for cT1a, presents results in favor of the more recent work [15,16,26,27,28,34], presented in our review.

Wośkowiak et al. [48] pointed out that data on long term follow-up of CA and RFA are limited. We managed to find results [16,34] on 10 year survival with mean OS, RFS, DSS, MFS of 72.95%, 80.65%, 97%, 100%, respectively for CA and 5 year [26,27,34] for RFA with mean OS, RFS, DSS, MFS of 87.8%, 95.8%, 98.13% and 97.07%, respectively, which are superior to the OS, RFS, DSS and MFS results of 75.8%, 93.5%, 97.9% and 87.7% in a study [50] mentioned in an earlier review [48].

The average MWA results for 1-year and 3-year survival, respectively, were: 96.67% and 96.8% for OS, 99.1 and 100% for MFS, 97.3% and 85.53% for RFS, 100% for DSS. Which compares with the 1 and 3 year results of older publications: Yu et al., 2014 [51], respectively OS = 97.9% and 89.7%, MFS = 97.9% and 87.4%, and Guan et al., 2012 [52], RFS = 100% and 95.1%, and for each group DSS = 100%, yielded similar results.

The least studied method is IRE, which despite being available for more than 20 years is still not very popular. The average results in our comparison were, respectively: 94.5%, 87%, 96%, 87% for 2-year OS, RFS, DSS and MFS. These results look worse than the 2-year OS, RFS, DSS, MFS at: 98.85%, 97.3%, 100%, 100% for MWA.

According to our findings CA seems to be the most studied method, followed by RFA and MWA, with a total cohort of 967, 621 and 495 patients, respectively. Also in the case of CA, we noted a large group of cT1b patients, which may indicate the use of this method in patients with more advanced cancer.

The new long-term data we compiled in the tables show a similarly favorable oncological outcome for RFA and CA. Which confirms the conclusions of Wośkowiak et al., regarding the efficacy of these methods.

In the case of MWA, the survival data from 6 articles mentioned in our review are limited to 3 years, therefore it seems difficult to compare this method to more extensively studied ones, as more long-term data are needed.

A clinical study [30] that retrospectively evaluated 297 patients with T1a RCC who underwent percutaneous ablation (navigated CT), performed with RF (82%), MWA (9%) or cryoablation (9%), was referenced to compare all thermoablative methods presented (excluding IRE). The average size of the tumor undergoing surgery was 2.4 cm, and the study cohort included populations that had been treated at the clinic over a 10-year period. The results showed that the success rate of the techniques was similar for all three methods, but primary efficacy at 1 month postablation was more likely to be achieved in the RF and MWA groups than cryoablation. Other values such as 2-year follow-up, RCC-related mortality, metastatic progression or local recurrence were equally common for each group. Also, eGFR did not differ between them. Thus, the authors concluded that both RF ablation, cryoablation and MWA after 2 years in the treatment of T1a RCC yield good (and equivalent between them) results in terms of therapeutic outcome, renal function and low rates of adverse events. For this reason, each modality can be used in patients who may benefit from their treatment. 

Having compared the 2017 AUA guidelines [53] with the latest 2021 ones [54], no changes were published regarding TA. For AS, we also found no significant changes in the recommendations.

For the EAU, the 2018 [55] and 2022 [7] guidelines for AS are no different except for the mention of a published paper [56]. In addition, they found no significant differences in 5-year CSS between AS and TA. In the case of TA, there is a lack of high-quality evidence to support the superiority of TA over PN. This method can only be recommended for ailing patients.

Changes to the NCCN guidelines between the 2018 [57] and 2022 [9] editions include more specific criteria for the use of TA and AS, while mentioning the possibility of needing repeat TA procedures to achieve a similar oncologic effect as with conventional surgery.

The latest ASCO guidelines were published in 2017 and there has been no update since then [4].

It seems that the progress achieved in terms of clinical results can be attributed to greater experience of operators, as well as the development of technology. 

The selection bias of retrospective work, although significant, should be borne in mind that randomized prospective studies involve much greater costs and time required. The goal for the next few years should be to establish diagnostic methods on comparable, large cohorts and also to establish predictors to help make the most tailored options for patients. And the collection of this information on a long-term scale, since the published data are promising but mostly based on short-term oncology studies.

## 5. Conclusions

According to current AUA guidelines, patients with SRM < 2 cm may undergo AS, TA and PN, and tailored treatment should take into account patient preference and the potential risks each method carries [5]. Despite the advances that have been made over the past few years in focal methods, the NCCN still recommends PN as the preferred method in patients with stage T1a tumors. The guidelines also recommend RN in selected patients, and leave active surveillance along with ablative techniques as available primary treatments [9]. AS is still recommended in sicker patients, especially the elderly, where surgery is high risk.

Several articles with long-term survival data (5–10 years) were collected, which can help evaluate the effectiveness of TA methods. We suppose that more long-term and larger cohort-based studies will help confirm the clinical utility of these methods and demonstrate their advantages over classical surgery.

In this article, we have updated the publication of Wośkowiak et al. [48], the data of TA procedures presented in our paper present results similar or better than the articles published before 2018. These conclusions were made on the basis of summary tables (Table 1 and Table 2), as well as analyzed guidelines and other reports from recent years.

## Figures and Tables

**Table 1 biomedicines-10-02583-t001:** Comparison of results for thermal ablation methods.

Method	Study	Study Group	Stage	Mean/Median Tumor Size (cm)	Mean/Median Follow-Up (Months)	Results		
Cryoablation	Morkos et al., 2020 [16]	134 patients	cT1a (115/134)	Median 2.8	Median 88.8	5 y OS = 87%, RFS = 85%, DSS = 94%	10 y OS = 72%, RFS = 69%, DSS = 94%,	
			cT1b (19/134)			OS = 88%, RFS = 89%, DSS = 94%	OS = 88%, RFS = 89%, DSS = 94%	
	Zangiacomo et al., 2021 [26]	69 patients	cT1a	Median 2.3	Mean 56	1 y OS = 100% PFS = 98.8% MFS = 100% DSS = 100%	5 y OS = 98.4%, PFS = 93% MFS = 100% DSS = 100%	PE= 95.7%
	Andrews et al., 2019 [27]	226 patients	cT1a (178/226)	Median 2.8	Median 75.6	5 y OS = 77% DSS = 100% RFS = 95.9% MFS = 100%		
			cT1b (48/226)	Median 4.8	Median 72	OS = 56% DSS = 91% RFS = 95% MFS = 90%		
	Spiliopoulos et al., 2021 [28]	53 patients (54 tumors)	cT1a (49/54)	Mean 2.8	Mean 46.7	1 y OS = 98% DFS = 100% PFS = 100% DSS = 100%	3 y OS = 90.3% DFS = 95.5% PFS = 94.3% DSS = 100%	5 y OS = 71.6% DFS = 88.6% PFS = 91% DSS = 95.8%
			cT1b (5/54)					
	Breen et al., 2018 [15]	220 patients (221 tumors)	cT1a (166/221)	Mean 3.4/Median 3.4	Median 31	3 y OS = 93.2% RFS = 97.2% MFS = 97.7%	5 y OS = 84.8% RFS = 93.9% MFS = 94.4%	
			cT1b (55/221)					
	Gunn et al., 2019 [29]	37 patients (37 tumors)	cT1b	Median 4.73	Mean 26.4	1 y RFS = 96.5% OS = 96.7% DSS = 100%	2 y RFS = 86.1% OS = 91.8% DSS = 100%	3 y RFS = 62.6% OS = 77.6% DSS = 100%
	Zhou et al., 2019 [30]	26 patients	cT1a	Mean 2.4	No data	2 y DFS = 100% PFS = 100% DSS = 100%	PE = 88%	
	Grange et al., 2019 [31]	23 patients	cT1b	Mean 4.56	Mean 13.9/Median 11	1 y PFS = 66.7% DSS = 100%	2 y PFS = 66.7% DSS = 85.7%	PE = 86.3% SE = 100%
	Shimizu et al., 2021 [32]	28 patients	cT1b	Mean 4.6	Mean 42	1 y OS = 96.3% DFS = 89.1% RFS = 92.7%	3 y OS = 92.3% DFS = 85.4% RFS = 92.7%	5 y OS = 89.1% DFS = 85.4% RFS = 92.7%
	UEMURA et al., 2021 [33]	48 patients	cT1a (46/48)	Median 2.6	Median 12	3 y RFS = 90.3% OS = 97.4%		
			cT1b (2/48)					
	Chan et al., 2022 [34]	103 patients	cT1a (72/103)	Median 2.85	Median 75.6	5 y DSS = 100% OS = 90.3% RFS = 98.5% MFS = 100%	10 y DSS = 100% OS = 73.9% RFS = 92.3% MFS = 100%	
			cT1b (31/103)	Median 4.5	Median 72.5	5 y DSS = 96.4% OS = 71% RFS = 92.8% MFS = 96.7%	10 y DSS = 96.4% OS = 43.5% RFS = 86.4% MFS = 96.7%	
Radiofrequency ablation	B. A. Johnson et al., 2019 [22]	106 patients (112 tumors)	cT1a	Mean 2.5	Median 79	10 y DFS = 81.5%, DSS = 94% MFS = 94% OS = 49%		
	Zangiacomo et al., 2021 [26]	16 patients	cT1a	Median 2.3	Mean 56	1 y OS = 100% PFS = 98.8% MFS = 100% DSS = 100%	5 y OS = 98.4%, PFS = 93% MFS = 100% DSS = 100%	PE = 95.7%
	Andrews et al., 2019 [27]	175 patients	cT1a	Median 1.9	Median 90	5 y OS = 72% DSS = 95.6% RFS = 95.9% MFS = 93.9%		
	Zhou et al., 2019 [30]	244 patients	cT1a	Mean 2.4	No data	2 y DFS = 100% PFS = 100% DSS = 100%	PE = 95%	
	Chan et al., 2022 [34]	100 patients	cT1a (87/100)	Median 2.8	Median 106	5 y DSS = 98.8% OS = 93% RFS = 95.7% MFS = 97.3%	10 y DSS = 98.8% OS = 89% RFS = 91.4% MFS = 97.3%	
			cT1b (13/100)	Median 4.5	Median 59.5	5 y DSS = 92.3% OS = 61.5% RFS = 87.5% MFS = 92.3%	10 y DSS = 92.3% OS = 52.8% RFS = 87.5% MFS = 92.3%	
Microwave ablation	Aarts et al., 2020 [23]	100 patients (108 tumors)	cT1a (77/100)	Median 2.8	Median 19	PE = 89% SE = 99%		
			cT1b (23/100)	Median 4.5		PE = 52% SE = 95%		
	Zhou et al., 2019 [30]	27 patients	cT1a	Mean 2.2	No data	2 y DFS = 100% PFS = 100% DSS = 100%	PE = 96%	
	Wilcox Vanden Berg et al., 2021 [25]	101 patients (110 tumors)	cT1a	Median 2.0	Median 12.5	1 y RFS = 97.3% OS = 100% DSS = 100% MFS = 100%	2 y RFS = 97.3% OS = 100% DSS = 100% MFS = 100%	PE = 98.2% SE = 100%
	Filippiadis et al., 2018 [24]	48 patients	cT1a (44/48)	Mean 3.1	Mean 43	3 year survival OS = 95.8% RFS = 73.75%		
			cT1b (4/48)					
	Guo and Arellano, 2021 [35]	106 patients (119 tumors)	cT1a	Mean 2.4	Median 24	1 y PFS = 100% OS = 99% DSS = 100%	2 y PFS = 92.8% OS = 97.7% DSS = 100%	3 y PFS = 90.6% OS = 94.6% DSS = 100%
	John et al., 2021 [36]	113 patients	cT1a (102/113)	Median 2.5	Median 12	1 y RFS = 97.3% MFS = 98.2% OS = 100%		
			cT1b (11/113)					
Irreversible electroporation	Wah et al., 2021 [37]	26 patients (30 tumors)	cT1a	Mean 2.5	Median 37	2 y RFS = 91% MFS = 87% DSS = 96% OS = 89%	3 y RFS = 91% MFS = 87% DSS = 96% OS = 89%	
	Canvasser et al., 2017 [38]	41 patients (42 tumors)	cT1a	Mean 2	Mean 22	2 y RFS = 83% OS = 100%		

DFS—disease-free survival, DSS—disease-specific survival, MFS—metastasis free survival, OS—overall survival, PE—primary efficacy, PFS—progression-free survival, RFS—recurrence-free survival, SE—secondary efficacy,. 1 y—1-year survival, 2 y—2-year survival, 3 y—3-year survival, 5 y—5-year survival, 10 y—10-year survival.

**Table 2 biomedicines-10-02583-t002:** Overview of mean results for ablative methods.

Method	No. of Studies	Study Group	Stage (Tumors)	Tumor Size (cm)		Follow-Up (Months)		Mean Results (%)								
				{No. of Studies with Included Results}												
				Mean	Median	Mean	Median	Follow-Up Time	OS	RFS	DSS	MFS	DFS	PFS	PE	SE
Cryoablation	11	967 patients (969 tumors)	cT1a (721)	2.87 {3}	2.79 {6}	37 {5}	40.37 {6}	1 year	99 {2}	n/a	100 {2}	100 {1}	100 {1}	99.4 {2}	91.85 {2}	n/a
								2 years	n/a	n/a	100 {1}	n/a	100 {1}	100 {1}		
								3 years	93.63 {3}	93.75 {2}	100 {1}	97.7 {1}	95.5 {1}	94.3 {1}		
								5 years	87.97 {6}	94.08 {4}	97.96 {5}	98.6 {4}	88.6 {1}	92 {2}		
								10 years	72.95 {2}	80.65 {2}	97 {2}	100 {1}	n/a	n/a		
			cT1b (248)	3.84 {4}	3.8 {6}			1 year	97 {3}	94.6 {2}	100 {3}	n/a	94.55 {2}	83.35 {2}	86.3 {1}	100 {1}
								2 years	91.8 {1}	86.1 {1}	92.85 {2}	n/a	n/a	66.7 {1}		
								3 years	90.16 {5}	85.7 {4}	100 {2}	97.7 {1}	90.45 {2}	94.3 {1}		
								5 years	77.78 {5}	92.68 {5}	93.8 {3}	93.7 {3}	85.4 {1}	n/a		
								10 years	65.75 {2}	87.7 {2}	95.2 {2}	96.7 {1}	n/a	n/a		
Radiofrequency ablation	5	621 patients (627 tumors)	cT1a (614)	2.5 {1}	2.35 {4}	56 {1}	83.92 {3}	1 year	100 {1}	n/a	100 {1}	100 {1}	n/a	98.8 {1}	95.35 {2}	n/a
								2 years	n/a	n/a	100 {1}	n/a	100 {1}	100 {1}		
								5 years	87.8 {3}	95.8 {2}	98.13 {3}	97.07 {3}	n/a	93 {1}		
								10 years	69 {2}	91.4 {1}	96.4 {2}	95.65 {2}	81.5 {1}	n/a		
			cT1b (13)	n/a	4.5 {1}			5 years	61.5 {1}	87.5 {1}	92.3 {1}	92.3 {1}	n/a	n/a	n/a	n/a
								10 years	52.8 {1}	87.5 {1}	92.3 {1}	92.3 {1}	n/a	n/a		
Microwave ablation	6	495 patients (525 tumors)	cT1a (479)	2.57 {3}	2.43 {3}	43 {1}	16.88 {4}	1 year	96.67 {3}	97.3 {2}	100 {2}	99.1 {2}	n/a	100 {1}	94.4 {3}	99.5 {2}
								2 years	98.85 {2}	97.3 {1}	100 {3}	100 {1}	100 {1}	96.4 {2}		
								3 years	96.8 {3}	85.53 {2}	100 {2}	100 {1}	n/a	90.6 {1}		
			cT1b (38)	3.1 {1}	3.5 {2}			1 year	100 {1}	97.3 {1}	n/a	98.2 {1}	n/a	n/a	52 {1}	95 {1}
								3 years	95.8 {1}	73.75 {1}	n/a	n/a	n/a	n/a		
Irreversible electroporation	2	67 (72 tumors)	cT1a (72)	2.25 {2}	n/a	22 {1}	37 {1}	2 years	94.5 {2}	87 {2}	96 {1}	87 {1}	n/a	n/a	n/a	n/a

DFS—disease-free survival, DSS—disease-specific survival, MFS—metastasis-free survival, OS—overall survival, PE—primary efficacy, PFS—progression-free survival, RFS—recurrence-free survival, SE—secondary efficacy.

## Data Availability

Not applicable.

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
