# Peer review of "The Role of Focal Therapy and Active Surveillance for Small Renal Mass Therapy"

_biomedicines, 2022, doi:10.3390/biomedicines10102583_

Round 1
Reviewer 1 Report
Overall, interesting and well-written paper. There two changes I would recommend. First consider putting in a guide to all the acronyms. Not all are spelled out on first reference, and a table spelling them out would help the flow. second the lay out of the table makes it very difficult to read, may be better to format it as a landscape table.
Author Response
Dear Reviewer 1,
Thank you for your time and contributory opinion. We did change our manuscript according to your suggestions. We added the table with acronyms etc. (please see below).
Text deleted – lines 69 to 97.
Text deleted – lines 120 to 131.
Text deleted – lines 181 to 191.
Text deleted – lines 257 to 264.
Text deleted – lines 310 to 326.
Text deleted – lines 331 to 340.
Text added – lines 351 to 353.
Text added – lines 384 to 389.
Text deleted – lines 390 to 393.
Text added – lines 403 to 412.
Text added – lines 428 to 441.
Text added – lines 459 to 462.
Added abbreviation table
Table 1
Various spaces deleted to make the table more compact
X year survival changed to abbreviation and added explainations to table legend
Deleted bold font - Morkos et al., 2020, Cryoablation, Method, Study
Table 2
Various changes implemented to make the table easier to understand
AS |
Active surveillance |
ASA |
American Society of Anesthesiology |
ASCO |
American Society of Clinical Oncology |
AUA |
American Urological Association |
CA |
Cryoablation |
CT |
Computed tomography |
DFS |
Disease-free survival |
DISSRM |
Delayed Intervention and Surveillance of Small Kidney Masses |
DSS |
Disease-specific survival |
EAU |
European Association of Urology |
eGFR |
Estimated glomer |
FDA |
Food and Drug Administration |
FT |
Focal therapy |
IRE |
Irreversible electroporation |
MFS |
Metastasis-free survival |
MWA |
Microwave ablation |
NCCN |
National Comprehensive Cancer Network |
OS |
Overall survival |
PE |
Primary efficacy |
PFS |
Progression-free survival |
PN |
Partial nephrectomy |
RCC |
Renal cell carcinoma |
RFA |
Radiofrequency ablation |
RFS |
Recurrence-free survival |
RN |
Radical nephrectomy |
SE |
Secondary efficacy |
SOC |
Standard of care |
SRM |
Small renal masses |
TA |
Thermal ablation |
Reviewer 2 Report
The article by Matuszczak titled “The role of focal therapy and active surveillance for small renal masses therapy” is a good summary of the current clinical research on focal therapy and active surveillance on SRMs, especially since 2018. The reviewer finds table 1 and table 2 very informative.
However, the reviewer would like to point out a few shortcomings and a few suggestions, mostly to de-emphasize the contents that have been covered extensively by other reviewers and expand more on the original contribution of this article, which is to cover the latest development since 2018.
1. A large amount of information is redundant from the review by Wośkowiak et al.
· Wośkowiak, P., Lewicka, K., Bureta, A., & Salagierski, M. (2019). Active surveillance and focal ablation for small renal masses: a better solution for comorbid patients. Archives of Medical Science, 16(5), 1111-1118.
Different focal therapy and active surveillance has been discussed extensively by numerous good reviews including the reviews and perspectives from Salagierski et al. Together, the introduction of various focal therapies and active surveillance take up more than ½ of the entire article, which the reviewer does not think is the focus of this review.
· Salagierski, M., Wojciechowska, A., Zając, K., Klatte, T., Thompson, R. H., Cadeddu, J. A., ... & Capitanio, U. (2018). The role of ablation and minimally invasive techniques in the management of small renal masses. European Urology Oncology, 1(5), 395-402.
· Kriegmair, M. C., Bertolo, R., Karakiewicz, P. I., Leibovich, B. C., Ljungberg, B., Mir, M. C., ... & Capitanio, U. (2018). Systematic review of the management of local kidney cancer relapse. European Urology Oncology, 1(6), 512-523.
· Mir, M. C., Capitanio, U., Bertolo, R., Ouzaid, I., Salagierski, M., Kriegmair, M., ... & Pierorazio, P. M. (2018). Role of active surveillance for localized small renal masses. European Urology Oncology, 1(3), 177-187.
The reviewer would recommend downsizing these parts and focus on my next recommendation.
2. The reviewer highly encourage the authors to compare this review, especially the findings presented in table 1 and 2, to the summary presented by Wośkowiak et al., to show since 2018, for example:
· What is new?
· What has changed?
· What may lead to changes to perception towards focal therapy and active surveillance?
· What are the new updates on the NCCN Clinical Practice Guidelines, AUA/EAU Guideline, ASCO recommendation that reflect the changes?
For this part, an expansion on the discussion on table 1 and table 2, together with an extended summary in section 4 would be helpful.
3. other minor comments
§ Some people may not agree that cryoablation is a form of thermal ablation, as the word “thermal” literally means “relating to heat.”
§ Table 1 appears to be fairly long. Not sure if the table can be made more compact, for example within a page?
Author Response
Dear Reviewer 2,
Thank you for your time and contributory opinion. We did change our manuscript according to your suggestions. We added the table with acronyms etc. (please see below). Hope the information given would be sufficient.
- The redundant parts of our manuscripts was transformed and partially deleted.
- The updates and the current guidelines were updated. Thanks for this idea.
- We did change some minor aspects and information in the tables.
Furthermore,
Text deleted – lines 69 to 97.
Text deleted – lines 120 to 131.
Text deleted – lines 181 to 191.
Text deleted – lines 257 to 264.
Text deleted – lines 310 to 326.
Text deleted – lines 331 to 340.
Text added – lines 351 to 353.
Text added – lines 384 to 389.
Text deleted – lines 390 to 393.
Text added – lines 403 to 412.
Text added – lines 428 to 441.
Text added – lines 459 to 462.
Added abbreviation table
Table 1
Various spaces deleted to make the table more compact
X year survival changed to abbreviation and added explainations to table legend
Deleted bold font - Morkos et al., 2020, Cryoablation, Method, Study
Table 2
Various changes implemented to make the table easier to understand
AS |
Active surveillance |
ASA |
American Society of Anesthesiology |
ASCO |
American Society of Clinical Oncology |
AUA |
American Urological Association |
CA |
Cryoablation |
CT |
Computed tomography |
DFS |
Disease-free survival |
DISSRM |
Delayed Intervention and Surveillance of Small Kidney Masses |
DSS |
Disease-specific survival |
EAU |
European Association of Urology |
eGFR |
Estimated glomer |
FDA |
Food and Drug Administration |
FT |
Focal therapy |
IRE |
Irreversible electroporation |
MFS |
Metastasis-free survival |
MWA |
Microwave ablation |
NCCN |
National Comprehensive Cancer Network |
OS |
Overall survival |
PE |
Primary efficacy |
PFS |
Progression-free survival |
PN |
Partial nephrectomy |
RCC |
Renal cell carcinoma |
RFA |
Radiofrequency ablation |
RFS |
Recurrence-free survival |
RN |
Radical nephrectomy |
SE |
Secondary efficacy |
SOC |
Standard of care |
SRM |
Small renal masses |
TA |
Thermal ablation |